# Patterns for Visual Management in Industry 4.0

**DOI:** 10.3390/s21196440

**Published:** 2021-09-27

**Authors:** Giuseppe Fenza, Vincenzo Loia, Giancarlo Nota

**Affiliations:** Dipartimento di Scienze Aziendali—Management & Innovation Systems (DISA-MIS), University of Salerno, 84084 Fisciano, Italy; gfenza@unisa.it (G.F.); loia@unisa.it (V.L.)

**Keywords:** Industry 4.0, human-computer interaction, visual management, visual patterns

## Abstract

The technologies of Industry 4.0 provide an opportunity to improve the effectiveness of Visual Management in manufacturing. The opportunity of improvement is twofold. From one side, Visual Management theory and practice can inspire the design of new software tools suitable for Industry 4.0; on the other side, the technology of Industry 4.0 can be used to increase the effectiveness of visual software tools. The paper first explores how the theoretical result on Visual Management can be used as a guideline to improve human-computer interaction, then a methodology is proposed for the design of visual patterns for manufacturing. Four visual patterns are presented that contribute to the solution of problems frequently encountered in discrete manufacturing industries; these patterns help to solve planning and control problems thus providing support to various management functions. Positive implications of this research concern people engagement and empowerment as well as improved problem solving, decision-making and management of manufacturing processes.

## 1. Introduction

Starting from the introduction of the Toyota Production System, developed in Toyota in the middle of the 20th century as a lean production management tool [1,2], the studies of Visual Management (VM) received an important boost [3]. In manufacturing, VM is made possible by communication devices able to return data about the state of a workplace, a machine tool, a group of machine tools (e.g., a cell or a production line), a production process, a department or the whole factory.

Traditionally, the *visual workplace* has been implemented by means of the 5S method that comprises five steps that aim at workers engagement to achieve productivity improvement and safety [4]. Well known VM systems for production planning and control (PPC) are:-the *kanban system* [5] for the just-in-time control of repetitive operations in manufacturing. A Kanban system uses two types of cards, production kanbans and transport kanbans, that provide visual evidence of what can be produced by a workstation or moved from one workstation to another.-*Communication Board* [6], such as an *andon board* that is placed in the proximity of a workstation or above a production line to provide data about the operating status of the equipment.

The benefits of VM have been acknowledged in several application domains. As reported in [7], VM systems used in manufacturing industries facilitate performance measurement and review, enable people engagement, improve internal and external communication, enhance collaboration and integration, support the development of a continuous improvement culture and foster innovation. VM has been applied also in transportation [8], construction [9,10] and health [11].

The VM discipline has developed with the contribution of professionals in the lean production field and is a widespread practice. Even if the success stories witness the usefulness of VM systems, scholars have argued that there is a lack of theoretical and conceptual understanding based on visual studies. [12,13,14]. On the one hand, visual and design research are rare within studies of management control systems and there is a need to perform research that takes into account the role and function of visual communication and the design of VM systems [15]. On the other hand, opportunities for improvement in visual factory management also arise from the advent of Industry 4.0 technologies that make it possible to rethink traditional VM applications.

The goal of this research is twofold. We first build on the VM theory so that *visual patterns* can be identified for industrial software applications. Then, we reverse the point of view trying to take advantage of the Industry 4.0 technologies and practice to improve the effectiveness of VM software tools.

This paper proposes some visual patterns for Industry 4.0 that can be taken as reference for the development of Graphical User Interfaces (GUI) or Human Machine Interfaces (HMI) for many kinds of software applications in the Industry 4.0 scenario. The research activities presented in this paper were carried out within the PICO & PRO project funded by the Italian Ministry of Economic Development. The PICO & PRO project saw the participation of three Italian universities and six companies operating in the automotive sector. Regarding the application of visual patterns for Industry 4.0 the expected benefits range from the easier design of software tools for manufacturing to the reusability of solutions already verified as best practices. Furthermore, the use of Industry 4.0 technologies in VM tools will render the data collection from different data sources much easier, so that selected data can qualify the notion of the production state in a precise and detailed way. An increased people engagement and empowerment as well as improved problem-solving, decision-making and performance of manufacturing processes are benefits that can be achieved by applying this research in manufacturing.

The paper is organized as follows. The examination of the literature in Section 2 reviews the fundamental aspects related to production facilities, classification of production systems and VM necessary for the development of this research. A model of a cyber-physical system that can be used as a reference for the design of visual software tools that operate in the Industry 4.0 is presented in Section 3. Starting from this model, a methodology for the design of software tools for VM, based on the idea of visual patterns for manufacturing, is presented in Section 4. Four visual patterns for designing and implementing visual software tools to solve recurring manufacturing problems are discussed in Section 5. Finally, a case study shows the implementation of the people evaluation pattern for an industry that participated in the PICO & PRO project. A discussion on the benefits of this research, its limitations and possible future developments closes the paper.

## 2. Literature Review

A production system can be defined as a collection of people, equipment and procedures organized to perform the manufacturing operations of a company [16]. Two fundamental components of a production system are facilities and manufacturing support systems; people working in the manufacturing support systems adopt methodologies and tools to operate the production facilities by means of planning, execution and control activities so that the expected production performances can be achieved or improved. Various ways of classifying production systems can be found in the production literature which depend on the purpose for which this classification is made. Typically, two important aspects are considered: the reference market (whose demand is expressed in terms of the quantity of goods to be produced, delivery times and flexibility) and the technologies that make production possible and convenient in a given context. In the next section, the structure of production facilities is briefly reviewed. Then, we look at two categories of production systems that appeared in the industry literature, the first introduced by Wortmann [17] and the second by Brandolese et al. [18]. These categories also outline some dynamic aspects of production processes and how they can activate the underlying structures. The awareness of structures, categories of production processes and their fundamental dynamics allow stating the context necessary to design patterns for VM systems.

### 2.1. Structure of Production Facilities

One of the most used models to represent the structures of manufacturing industries is the ISA 95 model, also known as the IEC/ISO 62264-1 standard. Although it has already been widespread since 1995, interest in this standard is constantly growing and its content is still relevant and valid. In the version of 2018 [19], the standard specifies conceptual interface content exchanged between manufacturing control functions and other enterprise functions. From a structural point of view, ISA95 consists of five levels (work units, work centers, area, site, enterprise) ranging from the physical elements that make the individual devices to the enterprise level; we can also consider the supra-system “supply chain” extending the traditional schema:*Device/unit process*: Individual device or machine tool in the manufacturing system that is performing a unit process [20].*Line/cell/multi-machine system*: A group of machines organized in a line layout (multiple workstations arranged in sequence where parts or assemblies are physically moved through the sequence to complete the product) or cellular layout (consisting of several workstations or machines designed to produce a limited variety of part configurations, specializing in the production of a given set of similar parts or products) [16].*Facility*: The relative location of equipment and/or work centers on the factory floor [21].*Multi-factory system*: Different facilities whose proximity to one another allows them to use possible synergies in terms of reuse of waste and lost energy streams [22].*Enterprise/global supply chain*: The flow and transformation of goods (as well as the flow of the associated information) from the raw materials stage to the end-user, including the supplier’s supplier and the customer’s customer. This flow of goods and information may encompass several different facilities (plants, warehouses, sales and distribution centers) belonging to several different business entities located in various parts of the globe [21].

### 2.2. Classification and Dynamics of Production Systems

Production systems can be classified from different points of view depending on the objectives for which the classification is proposed. It is also possible that a production system may belong to more than one classification system according to the point of view that the observer decides to adopt.

Among the first proposals for the classification of production systems, the one by [17] divides the plants based on the *customer decoupling point* which specifies the moment in which production goes from being based on forecast (make to stock) to being based on customer orders that triggers the activities of product design and manufacturing (engineer to order).

-*Make to Stock:* The industry produces inventories based on sales forecasts.-*Assembly to order*: The product assembly begins upon receipt of the order starting from manufactured or purchased components.-*Make to order*: Manufacturing industries make the product only after they have received the order. For the manufacturing processes that belong to this category, the product design activities are anticipated with respect to the time of order acquisition.-*Engineer to order*: As in the make to order modality, the product is made only when the order is received. However, the product design is necessary because the customer specification has unique features that require design and engineering activities before the manufacturing process can be triggered.

Another well-known classification schema is due to [18] that considers three analysis dimensions for production processes: (1) the market that specifies how the customer demand is formed and met by the manufacturing industry; (2) the technology adopted for discrete manufacturing (construction or assembly of parts) as well as the technology necessary for process industry; (3) management aspects that are devoted to the production of a single product, a batch of products (where the materials are processed in finite amounts or quantities over a finite period of time using a given equipment configuration) or a continuous production process where the production equipment is used exclusively for a given product. In Figure 1a the relationship between the type of production processes is shown. Figure 1b outlines how the three dimensions can be related. For example, if we consider an automotive industry that produces mechanical molds, then the typical demand is a single order for constructing a unity of product (the mold) with the manufacturing process that comprises both the construction process of parts and the assembly of MAKE or BUY components provided by external suppliers. Another example is a food industry that can produce either for the warehouse (based on a forecast) or according to a repetitive order modality if the production output is devoted to large-scale retailers.

The production classification system described in [23] is based on key components and properties of alternative production systems also considering Flexible Manufacturing Systems and Computer-Integrated Manufacturing. More recently, starting from a review of existing taxonomies, Sorensen et al. [24] proposed a classification scheme that helps ensure consistency during mapping of existing production systems, and assists in providing an overview of when, where and how fundamental functions of a production system are realized.

### 2.3. Visual Management

Many authors agree that VM is a discipline that has developed within the framework of the studies on the Toyota production system and Lean Management [3,10,25]. A simple definition of VM is reported in [26]:


*Visual Management is a management system that attempts to improve performance of an organisation by means of visual stimuli. These visual stimuli communicate important information of the organization at a glance, helping to convey relevant, easy to understand information in context.*


In other words, VM is a management system that lives within a context (a set of suprasystems made of production facilities, production process, suppliers, clients, etc.) with which VM interacts. The visual stimuli generated by VM convey relevant information toward the context where a worker (a part of the context) is solicited by dynamic interactions with VM. The worker that receives the stimuli is enabled to decide in a few seconds whether or not to activate a feedback action.

A similar definition is due to [27]:


*Visual Management can be defined as a management system that attempts to improve organizational performance through connecting and aligning organizational vision, core values, goals and culture with other management systems, work processes, workplace elements, and stakeholders, by means of stimuli, which directly address one or more of the five human senses (sight, hearing, feeling, smell and taste).*


The additional aspects of VM like organizational vision, core values, goal and culture and their alignment with other entities are considered in this definition. These aspects are relevant because they can contribute to improving production performance and safety. The five senses are also indicated as targets of stimuli generated by VM to underline ways other than sight to communicate messages.

A key feature of VM is the achievement of organizational process transparency [28] VM makes evident information about the production state in terms of data concerning the order release, scheduling and progress, as well as process deviations, plant layout, equipment status, etc. In manufacturing, this information is usually managed by software systems known as MES (Manufacturing Execution Systems) but VM comes into play when the information to the workers in the shop floor must be presented in the simplest possible way so that the state of the production context is perceptible in a few seconds and any feedback actions can be taken immediately.

Two fundamental concepts of VM are *visuals*, and *visual communication*. According to [13], visuals are made of different types of tables, diagrams or strategic lines on the shop floor. Visuals can include photos, drawing, charts, color surfaces, graphic forms and typefaces [29]. The point of view of [30] also considers a variety of forms, including pictures, graphs, film, web pages and architecture as parts of visuals

The combination of these points of view provides a characterization of visuals. However, it is worthwhile to observe that, even if visuals are usually adopted when we realize and use IT applications, digitalization is not convenient or is inapplicable in many scenarios where traditional solutions are preferred (e.g., accident prevention signs on a construction site or an iconic representation that suggest where to place mechanical tools on shadow boards). Therefore, the elements of visuals mentioned by Bell and Davidson make sense for IT applications only.

The communication act for which information is exchanged between individuals through a common system of symbols, signs or behavior receives a particular interpretation when the communication becomes visual in a working context such as manufacturing or construction. Greif [31] emphasizes one fundamental difference of visual communication with respect to traditional communication; visual communication implies that a visual message (that comprises visuals) is observed by everyone working in the considered area, everyone who passes through the area and who comes into the range of visibility. Furthermore, the visual message is intended for a group not just an individual and the observation is not enough because it is required that the meaning be understood. In the visual workplace approach [32], the communication happens by means of *visual devices*. This mechanism is intentionally designed to share information vital to the task so that what is supposed to happen does happen. The message must be perceived at a glance so that places become self-explanatory, self-ordering, self-regulating, and self-improving [33]. The following Table 1 resumes the type of visuals that convey different message semantics for the visual workplace.

In the literature synthesis on VM in production management [34], the authors remark that this discipline has originated and evolved through a set of distributed and somehow unconnected efforts, mainly by practitioners who generally focus on a more practical vision (how), rather than an in-depth more conceptual one (what). The lack of a unifying theory is reflected in the proposed terminology where VM, visual workplace, visual controls, visual factory, shop floor management, visual tools and visual communication in most cases denote the same thing: a visual approach to the management of a factory or a workplace. In the following we shall refer to the terms VM, according to the definition in [27], and to *visual control* (VC) as discussed in [35]. VC is one of the 14 principles of the Toyota Production System that highlights how it supports people in decision-making and problem-solving. The principle of VC is simple: given a work context, its state must be perceptible by observing the elements that belong to the context. If something deviates from the expected state, this fact should be recognizable at a glance so that a feedback action can start as soon as possible.

Since production planning and control is an important function of manufacturing and can receive support from VM and VC, the distinction between them is necessary because visual software tools are not limited to VC. VM is wider than VC; it is a management technique that includes VC but makes available other functions such as planning, coordinating and administering tasks to achieve a goal.

An overview of the functions of VM can be found in [36]. Apart from the already mentioned function of transparency, other VM functions are management by facts, job facilitation and discipline, among others.

Few attempts have been made to propose a theory for VM. *Affordance theory* has been used by [12] to explain why the design and use of visual devices works for operation management practice. Affordance is defined in [37] as “what the environment provides or furnishes for a human or an animal”; in VM it concerns the quality or property of an object that provides information on how it can or should be used. The essential feature of affordance is that physical structures within the environment are perceived directly by actors without any intermediate, conscious, cognitive processing. In other words, it is an opportunity for immediate action made possible both by the effectiveness of the actor and by the structures in the environment.

*Visual rhetoric* has been proposed by [30]. It is inspired by the classic study on Aristotelian rhetoric: the art of explaining, convincing and persuading someone about something, including the means of argumentation. A visual and rhetorical perspective on management control systems is discussed in a recent study by [15] where the persuasive purpose of lean boards, as well as the metaphoric and persuasive functions of the visuals and information design in management control systems, is discussed.

Another possible approach to the VM design in manufacturing is the *Situational Awareness* (SA) theory. Even if a short summary cannot convey the full depth of this important theory, it is useful to recall its main characteristics. In their influential book Endsley and Jones [38] define SA as:


*SA is the perception of the elements in the environment within a volume of time and space, the comprehension of their meaning, and the projection of their status in the near future.*


Essentially, what SA suggests to us is that when we must do a particular job or wish to reach a goal, we need to be aware of our surroundings. The main aspects of the definition are divided into three separate levels that are part of the Endsley and Debra SA model in dynamic decision making:-Level 1. Perception of the elements in the environment-Level 2. Comprehension of the current situation-Level 3. Projection of future status

As the concepts of SA have been applied in several operational scenarios, they can be taken as reference to a deeper understanding of planning, execution and control operations in manufacturing.

Focusing on human-computer interfaces, *context-aware* studies [39,40,41], as well as *usability* [42] are areas of research that are usually considered to improve communication between people and machines so that the performance of the underlying systems and processes can, in turn, be improved.

## 3. Modeling A Cyber-Physical System to Support VM in Industry 4.0

In recent years, the possibility of collecting data and information on the production state has increased. This has led to new generations of software traditionally used in the industrial sector (e.g., MES, ERP, CAD/CAM/CIM) enriched with new digital services. However, the advent of Industry 4.0 poses new challenges and opportunities. According to Hermann, *Industry 4.0 is a collective term for technologies and concepts of value chain organization. Within the modular structured Smart Factories of Industry 4.0, cyber-physical systems monitor physical processes, create a virtual copy of the physical world, and make decentralised decisions. Over the internet of things, cyber-physical systems communicate and cooperate with humans in real-time.* Via *the internet of services, both internal and cross-organizational services are offered and utilized by participants of the value chain* [43].

A fundamental component for the realization of industrial systems compliant with the Industry 4.0 paradigm is the cyber-physical system [44]. The model of Figure 2 represents the basic features of a cyber-physical system that reflects Hermann’s definition and that we propose as a reference to facilitate the design of VM software tools in the Industry 4.0 scenario. The model is centered on the simple device/unit process (a machine tool Mi in Figure 2) but can be easily extended to represent other structures described in Section 2.1.

Digital twins have been used in manufacturing industries because of their capability to reproduce the state of a manufacturing system, its structural and dynamic properties. Each physical component or subsystem in the production system receives a digital representation, collecting and storing real-time data for easy understanding, learning and reasoning. There are a number of potential ways that a product, process, or system digital twin may be used at all levels of manufacturing (e.g., machine, cell, line, facility or supply chain) such as optimizing production planning and scheduling, minimizing the impact of equipment downtime or enabling virtual commissioning [45].

A significant part of the digital twin is constituted by the digital representation of the machine tools with the main data characterizing the structure and state of the machines. Table 2 shows an extract from the census of machine tools operating in one of the companies that participated in the PICO & PRO project; the extract is representative of a common scenario in metalworking industries. The data represented in Table 2 can be used as a source to feed VM support tools. For example, the data contained in the machine tool log file can be used for production control, while the data recoverable from the sensors can be used for condition-based maintenance. In Section 5.4 we will show a visual pattern that feeds from the data contained in the machine ledgers of the machine tools for planned maintenance purposes. Most of the digital data concerning the cyber-physical system is stored in tables managed by the digital twin. However, in some applications it may be convenient to keep the data under the direct control of the tool that implements a particular function. This is the case with the people evaluation tool discussed in the case study reported in Section 5.3 where the data concerning people, jobs, skills and evaluations are under the direct responsibility of the VM software tool devoted to the people evaluation.

Apart from the structural aspect shown in Figure 2, the systemic behavior is triggered by the PPC control logic that sends command messages to the worker/machine subsystem to execute a production order. During the operation, a worker can supervise the machine behavior by means of an HMI device where the machine and unit process states are shown. In the meantime, execution data are automatically collected by sensors, sent to the electronic control of Mi that forward them on the communication network in the direction of the digital twin that maintains the current situation about the production system. When the PPC function needs data from the digital twin they are acquired by the control logic and sent to the Factory VM software tool that is in charge to visualize the situation of the equipment in the factory. Visual information can also be redirected to VM boards located in the shop floor.

As the software subsystems of the proposed Cyber-Physical Production System (CPPS) continuously exchange data with each other and with the cloud, an important question is how secure communication can be achieved. This non-functional feature can be obtained with an implementation that considers aspects of different kinds such as:-the mapping of functional subsystems of Figure 2 on architectures specialized in secure communication such as the one described in [46] which is based on two enabling technologies for IIoT: Smart Spaces for intelligent data processing and Virtual Private LAN Services;-ensure that the implemented software pursues security practices as goals of IIoT [47]. For example, the software tool implemented for the pattern described in Section 5.4 uses a reliable Application Programming Interface and access control in IIoT networks.

## 4. A Design Methodology for VM Software Systems

Most VM software solutions are ad hoc as they have been designed based on professional knowledge [12] or current practice (how to do it) [8,36]; therefore, they lack a solid theoretical foundation and are difficult to reuse.

On the other hand, theoretical approaches, such as affordance and rhetorical theories, are useful for understanding some of the fundamental aspects of VM and for orienting thinking during the design of human-computer interfaces but have limited practical use. We can try to reduce the gap between VM systems designed as ad hoc solutions for the manufacturing industry and the theories proposed for the VM. The steps below are part of the methodology for designing VM software systems that can be developed starting from the idea of visual patterns in manufacturing:Context specificationIdentification of process, problem, and assigned function to solve itDesign or reuse of visual patternsData manipulationHuman Computer Interaction

The first step, context specification, refers to the representation of resources that participate in the production process. Depending on the kind of problem that the visual pattern is supposed to solve, the context can comprise elements such as industrial facilities (see Section 2.1), equipment, layout, workers, energy and the factory where the production process happen; when appropriate, even the supply chain can be considered as part of the context. The second step, identification of process, problem and function to solve it, is necessary to qualify the problem (in a given context) that the pattern can contribute to solve and the function to which the problem is assigned. According to the overview on classification and dynamics of production systems of Section 2.2, the kind of production process, (e.g., make to stock) is first considered, the problem to solve is formalized and the function devoted to the problem solving is pointed out. The function could be chosen among “production planning and control”, “quality control” and “professional maintenance” just to name a few. Examples of problems to solve are (a) determining labor and equipment resources required to meet the production plan during the next week (the capacity planning function of “production planning and control”) or (b) the assignment of emergency work orders to maintainers so that they can execute emergency maintenance as soon as possible (part of the function “professional maintenance”). It is important to observe that:(1)a production process, understood as a set of activities aimed at the realization of a product/service, focuses on the dynamical aspects of the underlying structure, that is, we can perceive it as a structure in action;(2)to be fully qualified, the context must consider both the structure of a production system and the state of its components which, in turn, depend on the execution of the production process. In other words, the state of a production system is a dynamic notion as a state evolves when events occur that determine its change.

After the completion of the first two steps, a visual pattern can be designed or chosen among a set of available design pattern for manufacturing according to:-the structural properties of the context and its dynamical aspects;-the representation of the problem to solve.

The step data manipulation provides information about appropriate algorithms that manipulate input data so that the “visual” aspect of the pattern is sustained by a control logic. In this way, the designer can reuse the pattern adapting it, if necessary, to a particular context. Finally, the techniques for human-computer interaction are adopted to improve the HMI. In the next section, we provide some examples of visual patterns proposed as general solutions for different kinds of problems frequently encountered in discrete manufacturing industries.

## 5. Patterns for Visual Management in Industry 4.0

This section introduces four visual patterns for manufacturing: product decomposition, job scheduling, people evaluation and maintenance planning. They are representative of how software tools for different management problems can be designed and implemented in Industry 4.0.

### 5.1. Product Decomposition

Product decomposition is a hierarchical pattern used during PPC to construct an assembled product. The pattern shown in Figure 3 is typical of the “engineer to order” where the production process is executed by the industry after the reception of a customer order. According to the classification of Brandolese [18], the production is unitary and comprises the macro phases “design” and “manufacturing”; in its turn, manufacturing can be further divided in processing and assembly. The pattern visually describes the product structure, made of subsystems, and the planned operational phases to build each subsystem. In this production modality, planning is usually done backward as the product delivery date and the main milestones of the project are constraints established by the customer. Therefore, durations of phases in the pattern are filled from right to left inserting the planned time interval [*p_start*, *p_end*] for each phase. When the production process is triggered, the control data input follows the timeline, and durations of phases [*a_start*, *a_end*] are inserted from left to right.

At the intersection of a row and a column we find the *schedule planning/control unit* which highlights the planned time interval [*p_start*, *p_end*] in which a phase f_j_ will be carried out on a subsystem S_i_. This time interval is compared with the actual execution time interval [*a_start*, *a_end*] to identify potential deviations from the production plan.

*Theoretical guidelines*. The pattern is inspired by the hierarchical decomposition approach of system theory used in system design. In fact, both the product and the production process are decomposed into low-level entities, subsystems and phases respectively. This pattern can be recursively applied to low-level entities. A generic subsystem *Si* can, for example, be detailed in its composing parts. The pattern can then be reused where the subsystems (first level decomposition) are replaced by components (second level decomposition) and the phases by operation on the components in order to build *Si.* Visuals adopted in this pattern are cells, rows, columns and the array of numbers for the calendar; eventually, photos can be adopted as soon as the subsystems are realized. A visual signal is also adopted for what concerns the schedule planning/control unit. Here, appropriate colors can be used to improve communication; for example, white, yellow and red cells mean respected deadline, slight delay and severe delay, respectively.

When a worker looks at this cell, he can observe any deviations from the planned schedule and can easily be convinced (visual rhetoric) of the need to initiate an immediate feedback action.

*Industry 4.0 technologies*. Execution data can be automatically collected by the underlying cyber-physical system. As shown in Figure 2, each machine tool is equipped with sensors, Programming Logic Computer or Numerical Computer Control, that are able to take over various kinds of execution data. In particular, while the input [*p_start*, *p_end*] is a task of the human being, the data [*a_start*, *a_end*] are automatically collected and stored within the digital twin where they can be retrieved for future monitoring and control activities.

### 5.2. Job Scheduling

Scheduling is a problem that must be solved to allocate resources to tasks over a given time period, for example, a day or a week. It is a well-known decision-making process frequently used in manufacturing to optimize a variable, for example to minimize the processing time of a set of jobs. The pattern job scheduling, shown in Figure 4, aims at providing to the decisor an easy-to-use GUI during the allocation of machine tools to tasks. Ease of use is an advisable characteristic of this pattern as the scheduling activity can repeat itself several times during a day due to events such as breakdown of a machine, unsatisfactory output quality, new demand for already allocated production resources, unavailable tools, etc. The pattern is divided into three sections. The first one shows the current orders backlog and allows the selection (underlined text) of production orders, parts to build, and the machines on which to execute the operations. In this example, the job list comprises all the necessary operations to build Part_ik_ and Part_j1_. The second section shows the schedule as computed by the scheduling algorithm. Finally, the third section receives data from the cyber-physical system about events that change the current context state. The event can be generated either by a worker that supervises a machine through an I/O device or automatically by the control logic of the involved machine.

*Theoretical guidelines*. In discrete manufacturing, a production order (job) must be transformed into operations necessary to produce the parts to be assembled into the finished product. Each job j consists of a set of Oij operations, where the generic Oij must be executed on machine i. Each Oij has an associated processing time tij. Furthermore, the operations are ordered to respect the sequence provided for obtaining a part. The scheduling algorithm depends on the structure of production facilities and the variable to optimize; therefore, it must be selected accordingly [48].

The design of this pattern has been guided by the SA theory. From the point of view of who uses the scheduling algorithm, the perception of the elements in the environment comprises the list of the backlog orders for the current day, the selected orders that can be executed with the available resources, the parts to build during the day and the machines in the shop floor necessary to produce the parts. The data that contribute to the comprehension of the situation are the input of the scheduling algorithm and the generated schedule. When the operations are triggered, events can happen in the environment that change the state derived from the execution of planned operations. This changed state modifies the perception of the decisor that can, based on stimuli received from the environment, decide to reschedule the jobs or not.

*Industry 4.0 technologies*. The cyber-physical system provides information useful to decide if a rescheduling is necessary. Data coming from machine sensors about conditions like machine breakdown, unavailable tools or poor output quality are automatically detected and redirected to the decisor.

### 5.3. People Evaluation

The pattern proposed here concerns the ability of people to execute assigned jobs. Input data to the pattern are people, jobs, necessary skills to execute the jobs and appraisals data. The output consists of appraisals expressed in graphical form and training needs. The pattern of Figure 5 can be applied to workers performing routine jobs whose different skills can be estimated by assigning numerical values, for example from 1 (poor skill) to 5 (excellent skill); furthermore, each job assumes expected values of skills necessary to perform it. A given worker receives a judgment from an assessor for the required skills as a list of values v1, … vn.

The awareness of the growth of a worker’s skills is made visible by the comparison between the previously estimated values and the latest ones. For example, the triple (expected, assessed, prev.value) = (5, 3, 2) for a skill certifies that the worker has acquired a point from the previous evaluation of 2 which is insufficient to perform a critical task. The radar chart makes immediately evident:the progress of the worker from the previous evaluation;the training requirement for each skill required to perform a job is defined by the difference between the expected value and the assessed value.

The job evaluation is a recurring task that can also be triggered when an observed anomaly during the production process is amenable to a worker. In this case, a Human Error Root Cause Analysis (HERCA) is performed trying to identify the possible cause of failure among a predefined list of root causes.

*Theoretical guidelines*. The Human Resource Management (HRM) literature has given and still gives attention to the management of human resources in various kinds of organizations. Scholars have studied HRM in small firms [49] and large companies [50,51], as well as studying perspectives of HRM that range from the project-oriented companies [52] to international HRM [51,53]. The importance of training and development in employee performance and evaluation is discussed in [54]. Again, the pattern can be seen with the lens of SA. In fact, the assessor sees a context made of people, jobs, skills and appraisals data. She evaluates a worker with respect to an assigned job trying to change the future state of the context establishing training needs for the evaluated worker.

*Industry 4.0 technologies*. The PE pattern can be implemented both as a traditional web application and as a software tool that interacts with the underlying cyber-physical system. In the first case, a simple application on a smartphone or tablet is sufficient to allow the assessor to periodically interview the worker for the purposes of his appraisal. In the second case, the use of the PE pattern could be triggered by the automatic signal concerning a production anomaly. A typical scenario is where the production manager receives an automatic anomaly signal from a worker/machine type system. If the analysis identifies the cause amenable to a human error, the signal is forwarded to the assessor to start a new appraisal cycle on the worker’s skills who caused the anomaly.

### 5.4. Maintenance Planning

For the manufacturing industry, equipment maintenance is critical for the management of the production process. Maintenance activities are motivated by the fact that the equipment is intended to last a long time and must be maintained to restore it to an acceptable operational condition. In large industries it is not rare to observe three different roles involved in operational activities for the equipment maintenance:(1)The *equipment responsible* that has the technical knowledge of a specific machine and can decide when the maintenance activities must be scheduled on which parts of the machine; it is assumed that each machine has a register, called *machine ledger*, that records the machine structure, the machine history (failures, maintenance activities, etc.) and the schedule of maintenance activities to be executed in the future. This role decides periodic maintenance, usually taking as reference a time horizon of one semester or one year.(2)The *maintenance planner* that receives the input from two sources, the machine ledgers for the equipment in the factory and the emergency work order list. Starting from this input, the planner schedules the maintenance activities weekly, deciding how the maintenance activities must be assigned to the maintainers. Eventually, the planner can reuse this pattern also on a daily basis when an emergency requires the immediate assignment of a maintenance order to a maintainer.(3)The *maintainer* that receives maintenance orders from the planner and executes the maintenance activities.

The maintenance planning pattern of Figure 6 can be used by a maintenance planner to schedule the periodic maintenance of machine tools. The pattern shows that in the chosen week (number 7 in the example) a mechanical activity in the storage area that lasts on average 60 min, must be assigned to a maintainer. A set of skills is required to do the job and the planner can provide further information like the description of the intervention and notes about the workplace. Finally, the planner can select a free maintainer from a list of available workers. The choice is made according to the competencies needed and the residual free time of a worker; after the job assignment, the percentage of free time to the worker is reduced accordingly.

*Theoretical guidelines*. Among the different approaches to maintenance [55], the time-based maintenance is a period-based maintenance policy, in which maintenance actions are carried out periodically with predetermined schedules [56]. In other words, it is a technique that involves scheduling maintenance activities. This means that time is the enabling factor, that is, the trigger of the activity. The approach is based on the knowledge of Mean Time Between Failures and Mean Time To Repair values to determine the most convenient period for this kind of maintenance.

*Industry 4.0 technologies*. The maintenance planning pattern takes input from both the machine ledgers and the emergency work order list. On one hand, planning activity can take advantage of cloud computing as the roles of equipment responsible, and the maintenance planner can share the data that enable them to choose the best maintenance decisions. On the other hand, there are situations where a machine tool is able to send an emergency signal to the maintenance planner about an emergency state that requires immediate action. A visual signal can then be sent to a maintainer that is free or that is executing a lower priority job so that the maintenance activity can start immediately.

## 6. Case Study

Although the visual patterns can be applied in small, medium and enterprise-level firms, the eventuality of their usage must be carefully evaluated. For example, small enterprises could decide to use the schedule pattern because it can be managed by one decisor; however, the same enterprise could not be able to use the maintenance planning pattern due to insufficient human and technological resources.

The case study discussed in this section shows the implementation of the pattern people evaluation in an automotive company operating in south Italy with more than 500 employees, 487 of them blue-collars, distributed over four production sites. In the words of the company CEO:

“We care about people. Our group creates an environment for its people to learn, share, grow, develop, inspire and be happy in their workplaces”.

An essential aspect related to this vision is the assessment of people’s knowledge so that learning paths can be identified pursuing learning goals. The realized software tool for the “people evaluation” reflects the corresponding visual pattern. The GUI shown in Figure 7 is part of a software application that allows the creation of a database of workers and tasks to execute on the shop floor together with the definition of necessary skills for each task. The implemented tool reflects faithfully the pattern of Figure 5. In fact, the assignment of a task to a worker allows us to see the set of necessary skills, and the corresponding (expected, assessed, prev. value) triples from which derive the worker’s learning needs. Figure 7 shows the situation where the employee Adamis Ara receives a task of electromechanical nature for which a set of skills is required; considering the skill PLC programming, the expected value is 5 while the assessed value is 1 remarking the need of a learning session. In the case of human error during the task execution, a list of possible causes can be shown allowing the evaluator to identify the root cause, for example, “1.1 Training has not been completed”. Moreover, the visual information can be filtered allowing to reduce the user’s cognitive overload. In the example, the user decided to filter the previous evaluation data by removing it from the radar chart view.

The current version of the people assessment tool has been implemented as a web application in a cloud environment to achieve the following features:(1)lean application, not necessarily dependent on the implementation of the CPPS;(2)access to data in places other than the factory.

In the extended version which involves interaction with the cyber-physical system, additional benefits can be obtained as described in Section 5.3. In this case, the tool interacts with the PPC control logic module (see Figure 2) which has, among other things, the task of retrieving the data kept by the digital twin of potential interest for VM software tools. In the scenario of people evaluation, a signal to be rendered visual relative to the identification of an anomaly attributable to a human being is redirected to the people evaluation tool so that a feedback action can be eventually triggered.

After one year of using this software application, we evaluated its benefits from two points of view: individual and collective. For the assessment of individual benefits, a questionnaire with some questions was proposed, inspired by the theoretical studies cited in the literature review, to 40 workers of whom 28 answered the questions, and a skills assessor. Table 3 reports the results of this qualitative assessment.

Collective benefits have been arranged with the collaboration of a manager of human resources. They look at the overall rating of the blue-collars and have been measured by means of simple database queries. In Table 2, the *expected work capacity* is the value of total credits for the skills of all the employees. This value is expected by the company to perform the jobs in the best possible way. Analogously, *total credits* express the current sum of all the evaluated skills for the employees as they have been assessed by a skill evaluator. At the time of the measurement, the expected work capacity and total credits were respectively 20,948 and 17,293, the difference between which shows a *training need* of 3655 credits. Table 4 shows some KPI developed during the case study.

## 7. Conclusions

Since the implementation of the Toyota Production System, the numerous applications developed in the industrial field testify that VM can support various business functions such as operations management, performance measurement, strategy development, etc. [6]. VM contributes to reducing both production times and production variability; this explains its central role in lean production [16,38].

Born from empirical studies, this discipline has shown its practical utility, but to date, it does not have the support of a theory capable of interpreting it in its various facets. This lack is probably due to the multidisciplinary nature of the VM, which requires knowledge in different areas such as design and implementation of human/machine interfaces, cognitive psychology, philosophy and graphic design. Applications in the industrial field also require knowledge of the application domain and Industry 4.0 technologies. The attempts of some scholars who have proposed theories such as visual rhetoric or which refer to the theory of affordance allow us to frame some general aspects relating to man/machine interaction but are of limited practical utility. Studies on the usability of the software offer principles and guidelines for the design of easy-to-use software systems [39] which, however, require in-depth analysis when applied to the manufacturing sector.

This paper contributes to reducing the gap between theoretical studies and the need for concrete solutions required in real application domains, introducing the idea of visual patterns for manufacturing.

The concept of pattern is not new and has been used in many disciplines, including architecture [57] and software engineering [58]. However, as far as we know, there is no evidence in the literature of the idea of visual patterns applied to Industry 4.0. In [59], a goal-oriented design is proposed as an approach to HMI that comprise the situational awareness and techniques for effective design elements but there is no mention of the concept of reusability which is one of the main advantages of patterns. Probably, the work that comes closest to the ideas presented here is presented by Hollifield [60] in which piping and instrumentation diagrams are used to maintain situational awareness to recognize abnormal situations or equipment malfunctions. Our approach to the design of VM systems takes advantage of Industry 4.0 technologies to improve existing solutions and allows us to overcome their limitations such as:-difficulty of data sharing between multiple roles;-poor GUI/HMI due to an unsatisfying context representation;-unavailability of data generated by the underlying cyber-physical system;-difficulty in implementing real-time reactive behaviors.

Four visual patterns, inspired by the SA theory, have been designed to show how software tools for VM can be boosted up by means of Industry 4.0 technologies. From one side, the benefits of visual patterns are similar to those already obtained from the design patterns widely used in object design (increase developer productivity, promote reuse of development efforts, describe proven solution to common problems, etc.) with an important difference: design patterns focus on the benefits that a software designer can obtain using them, visual patterns emphasize the human-machine interface trying to obtain a better performance of the machine/worker system and the entire production system. On the other side, the benefits deriving from the use of visual patterns for manufacturing are managerial in nature and in line with what has already been observed by other authors [7], that is, major people engagement and empowerment, improved problem solving, decision-making and management of manufacturing processes.

This study has some limitations. With the current state of knowledge, the evaluation of the performances deriving from the use of visual patterns is empirical and approximate, even if it can consider both qualitative and quantitative evaluations. From a qualitative assessment point of view, end-user interviews can provide insight into the perceived quality of using the software tool that implements the pattern. This approach was used for the case study discussed in the previous section and can be adopted whatever pattern we consider. Quantitative assessment could be obtained by comparing current VM systems with visual pattern-based VM systems. However, since industries are primarily concerned with the performance of production processes, data is rarely collected regarding the effectiveness of the interaction between humans and machines. In this scenario, for the quantitative evaluation of the effectiveness of VM systems, we have two possible approaches:(1)evaluate the effectiveness of VM in a simulated environment;(2)infer information based on the comparison between the overall performance of the production system before and after the introduction of the new VM system.

The second approach has been used for the KPI discussed in the case study. Due to the variety of structures and production processes resumed in Section 2.2, many other visual patterns remain to be identified and designed; the following list highlights some common problems in discrete manufacturing that could be solved with the contribution of visual patterns:-Resource capacity (analysis of what the industry is capable of producing versus the expected demand)-Route sheet (the workflow of manufacturing operations to be performed on a work part)-Production line (the performance of a group of machines organized in a line layout)-Andon board (representation of machine/group of machines with data about state, KPI, trends, etc.)-Quality control (to maintain the quality of production under statistical control).

One important problem to consider is pattern integration. An integrated system centered on the visual pattern is challenging as it requires to build on a coherence property that, given the variety of problems to which the patterns are addressed, is very difficult to achieve. However, there is one point of view that can contribute to the achievement of the coherence property that regards the influence that one problem can have on another. This aspect has been studied in the World Class Manufacturing methodology [61], where several influences have been identified and specified. For example, when an anomaly is detected on a produced part or product, an analyst can trace the cause of the problem to a defect in the machine tool. This cause can trigger a message toward the maintenance planner that, in his turn, redirects the emergency work order to a free maintainer. This situation can benefit from integrating different visual patterns and is a research topic that will be addressed in the near future.

## Figures and Tables

**Figure 1 sensors-21-06440-f001:**
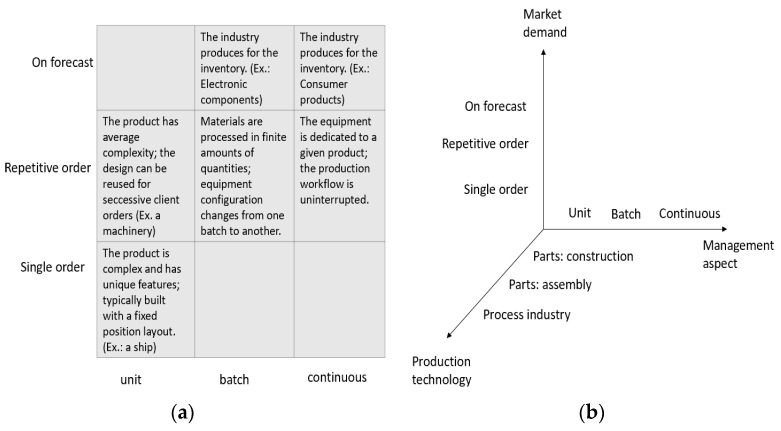
(**a**) Relationships between production types; (**b**) three-dimensional classification of production systems (Adapted from Brandolese).

**Figure 2 sensors-21-06440-f002:**
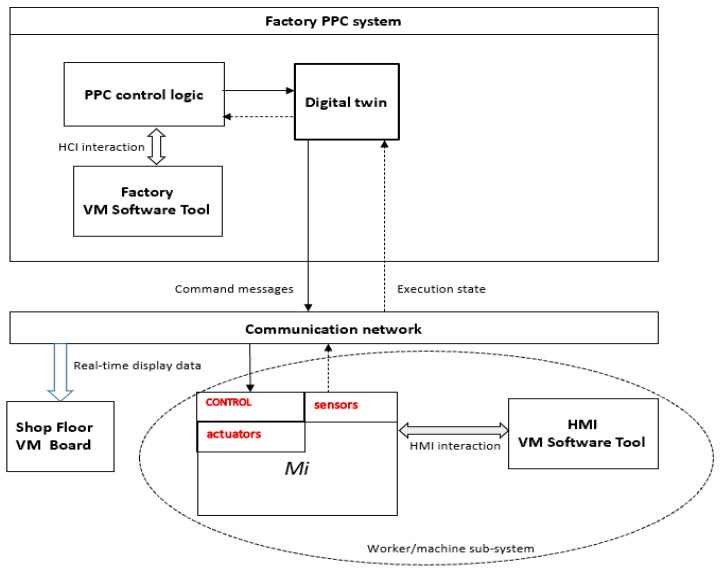
Model of a cyber-physical system with HMI, HCI and VM communication board.

**Figure 3 sensors-21-06440-f003:**
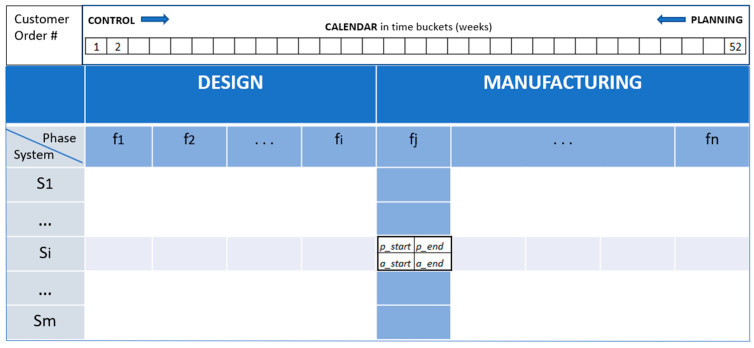
The visual pattern “product decomposition”.

**Figure 4 sensors-21-06440-f004:**
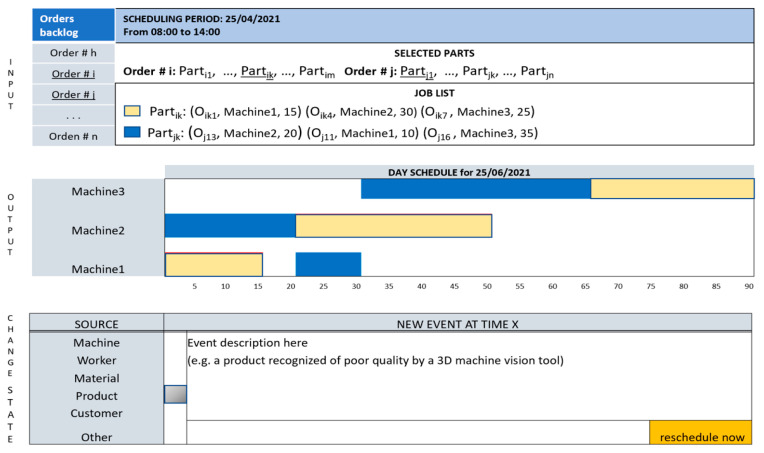
The pattern “Job scheduling”.

**Figure 5 sensors-21-06440-f005:**
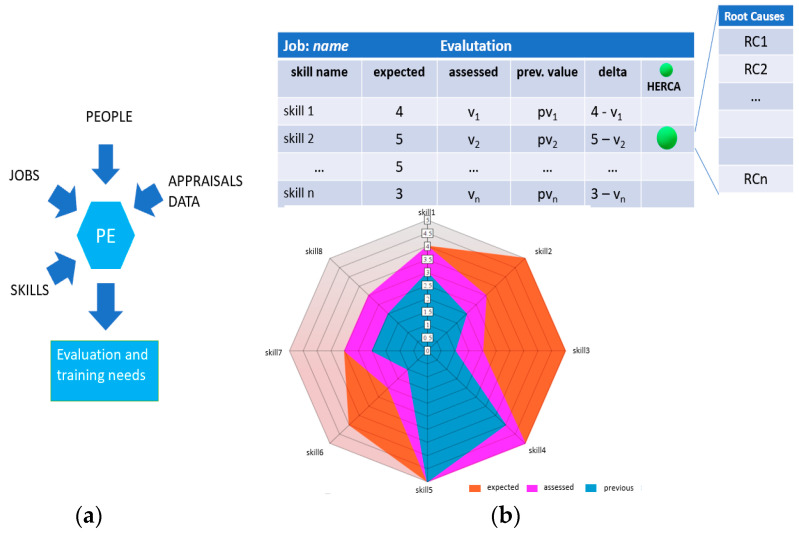
The pattern “People evaluation”. (**a**) Input and output of PE. (**b**) The evaluation of skills given a person and a job.

**Figure 6 sensors-21-06440-f006:**
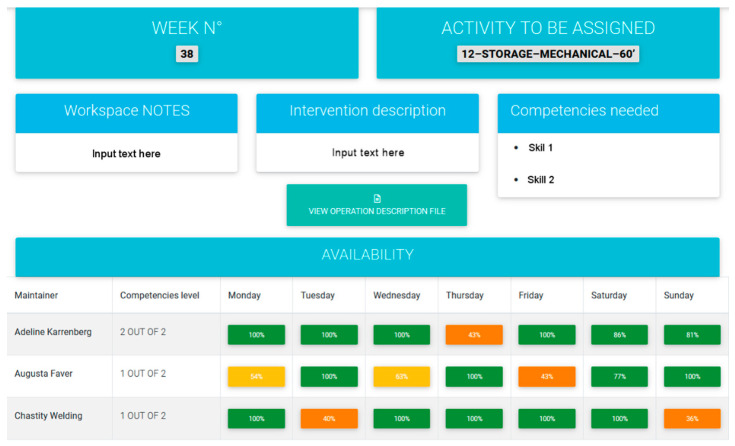
The pattern “Maintenance planning”.

**Figure 7 sensors-21-06440-f007:**
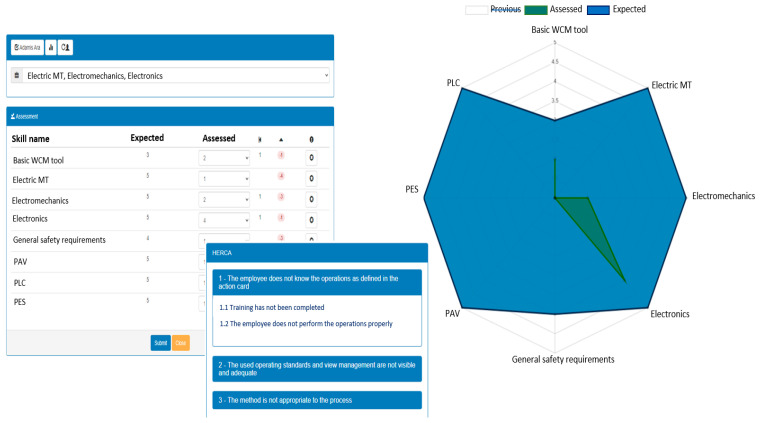
The GUI of the software tool for people evaluation.

**Table 1 sensors-21-06440-t001:** Message semantics of four visuals in visual workplaces realizations.

Visual Type	Message Semantics	Examples
Visual indicator	To provide or share information with a receiver that is not obliged to perform any action relied to the received message.	- Visuals that provide indications of safety exit doors from a shop floor
Visual signal	Provides a certain message to the receiver that takes attention. Visual signal expects the receiver reaction.	- Traffic lights - A visual on a machine tool display highlighting the need for a maintenance activity
Visual controls	The physical structure of the device sends the message and the response taken by the receiver is no longer limited solely by the message itself because using the device constrains potential future action. Limit and guide human actions.	- Parking lines - Kanban cards - Safety control using a protection grid from a production line
Visual guarantees	Allow only the desired outcome. Also known as mistake-proofing or poka-yoke.	- The visual state of a device or machine tool equipped with a vision system that discards output of poor quality

**Table 2 sensors-21-06440-t002:** Digital representation of machine tools.

Machine Code	Function	Machine Brand/Model	Year	CNC Brand/Model	PLC Brand/Model	CNC Address	PLC Address	Log File	Sensors	Machine Ledger
…										
ME3	Milling	MECOF/CS500	2015	Selca/S4045P		…		f3	Sensors ME3	ML-ME3
TE1	Milling	Tecmu	2016	Selca/3045		…		f4	Sensors TE1	ML-TE1
CH1	EDM Machine	Charmilles/510	2014	Charmilles		…		f5	Sensors CH1	ML-CH1
MO1	Press	Mossini/2000 Ton	1994		Siemens/S7/300		…	f6	Sensors MO1	ML-MO1
TR1	Laser cutting	Trumpf/TrueLaserCell7040	2018	Trumpf Op.Sys840D		…		f7	Sensors TR1	ML-TR1

**Table 3 sensors-21-06440-t003:** Assessment of the people evaluation pattern: individual benefits.

	Sample of 28 Blue-Collar				Evaluator
	inadequate	sufficient	fair	good	
1. Is the assessment of work skills correctly reported in the visual pattern?	2	6	15	5	fair
2. Is the need for training evident and immediately perceptible?	4	12	8	4	good
3. Is the GUI intuitive and easy to use?		7	10	11	good
4. Is information on the training gap to acquire new job skills an incentive to fill the gap as soon as possible?	1	6	12	9	sufficient
5. Does the root cause analysis help to improve the perception of the work context and the specific situations that led to the error?		14	9	5	fair

**Table 4 sensors-21-06440-t004:** Assessment of the people evaluation pattern: collective benefits.

KPI	Value
Expected work capacity	20,948
Total credits	17,293
Training need	−3655
Percentage of total credits compared to expected skills	82.55%
Percentage of training needs	17.55%
Percentage increase in learning sessions	12%
Number of credits gained in one year	727
Number of certifications acquired in one year	8
Percentage of reduction of quality problems due to human beings	9%
World Class Manufacturing compiance	high

## Data Availability

Not applicable.

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
