# Peer review of "Patterns for Visual Management in Industry 4.0"

_sensors, 2021, doi:10.3390/s21196440_

Round 1
Reviewer 1 Report
I think this paper is a solid work and is novel. There are many new ideas and concepts in the paper,
1 the industry 4.0 is a hot topic and the authors try to give some methods to bridge the gap between theory and practice, that is a good try.
2 the organization of the paper is well, especially the framework first, the four methods and then the case study, it is a solid work
But some suggestions:
1. Section 3 is important and thus it is needed to be detailed described, especially 2
2. There are two sections numbered with 4, please correct that
3. Case study can be more detailed given, for it maily supports your proposal of the paper
4. There are no comparision work, please add this part if there exists any.
Author Response
The reviewer’s comments are all relevant. We have used them to improve the paper quality as follow:
- Section 3 is important and thus it is needed to be detailed described, especially 2
Answer: As the digital twin is an important part of the cyber-physical model presented in fig. 2, we decided to detail this part by inserting a table that shows an extract from the census of machine tools operating in one of the companies that participated in the Pico&Pro project. Apart from the digital representation of the structure of a cyber-physical system in the digital twin, examples of the use of these data have been provided as well.
(lines 325-349).
- There are two sections numbered with 4, please correct that
Answer: done.
- The case study can be more detailed given, for it mainly supports your proposal of the paper.
Answer: The GUI of figure 7 has been explained in greater detail and the fundamental characteristics of the people evaluation tool have also been explained with respect to the model of figure 2 (lines 610-635).
- There are no comparision work, please add this part if there exists any.
Answer: Although the concept of pattern is well known, there is no evidence in the literature of the idea of visual patterns applied to industry 4.0. However, we inserted a couple of recent works that are close to the idea presented in our paper (lines 677-685).

Reviewer 2 Report
In order to be in line with the template for writing a paper, adjustments are needed:
- All References should be corrected according to the instructions as well as the citations in the text
- Some sources are not mentioned in the text,
- incorrect numbering of individual chapters and subchapters:
4 chapters - lines 345 and 400
Subchapter 4.2 - lines 446 and 530
- correct the numbering of individual subchapters
- adjust the description of Figures 3, 4 and 7 according to the template
- check the correctness of the term - r. 33 - "kamban system"
The authors could add a brief description of the software applications used in the presented case study.
Also compare your work to others in last five year.
Author Response
The reviewer’s comments are all relevant. We have used them to improve the paper quality as follow:
- All References should be corrected according to the instructions as well as the citations in the text
Answer: done
- Some sources are not mentioned in the text
Answer: done
- incorrect numbering of individual chapters and subchapters:
4 chapters - lines 345 and 400
Subchapter 4.2 - lines 446 and 530
- correct the numbering of individual subchapters
Answer: done
- adjust the description of Figures 3, 4 and 7 according to the template
Answer: done.
- check the correctness of the term - r. 33 - "kamban system"
Answer: done.
- The authors could add a brief description of the software applications used in the presented case study.
Answer: The GUI of figure 7 has been explained in greater detail and the fundamental characteristics of the people evaluation tool have also been explained with respect to the model of figure 2 (lines 610-635).
- Also compare your work to others in last five year.
Answer: Although the concept of pattern is well known, there is no evidence in the literature of the idea of visual patterns applied to industry 4.0. However, we inserted a couple of recent works that are close to the idea presented in our paper (lines 677-685).
